# Providing Oral Healthcare to Older Patients—Do We Have What It Takes?

**DOI:** 10.3390/ijerph20136234

**Published:** 2023-06-27

**Authors:** Cristiane da Mata, Patrick Finbarr Allen

**Affiliations:** 1Department of Restorative Dentistry, Cork Dental School and Hospital, University College Cork, T12 E8YV Cork, Ireland; 2Oral Health Services Research Centre, University College Cork, T12 E8YV Cork, Ireland; f.allen@ucc.ie

**Keywords:** gerodontology, dental training, barriers to oral healthcare, frailty, dental status

## Abstract

Over the past decades, there has been an increase in the number of natural teeth that are maintained into older age, and this has represented an increase in the need for more complex dental treatment for this population. A trained workforce is needed in order to provide dental care to the different groups of elderly. Currently, the undergraduate training in gerodontology offered by dental schools seems to be limited, with great variation among dental schools worldwide. Given the heterogeneity of elderly groups, it is unlikely that new graduates from dental schools can be deemed competent to deal with the different groups of elderly. In this article, barriers to oral healthcare’s provision to older adults are discussed, including the lack of appropriately trained dental professionals. Training pathways are discussed, including the preparation of undergraduate education to provide a suitable foundation to be developed further in postgraduate education. It is also proposed that older adults are classified according to their dependency level and each level is managed by properly trained dental professionals. In order to upskill general dental practitioners to care for these patients, postgraduate certification programmes could be structured to provide additional training. Furthermore, the development of geriatric oral health educational programmes for non-dental healthcare workers is recommended.

## 1. Introduction

The ageing of the global population is the most important medical and social demographic problem worldwide, according to the WHO [1]. The share of people over 65 in Europe is expected to increase from today’s 20% to 30% by 2050. In North America, the equivalent estimates are 13% and 22% [2]. In Asia, the statistics are even more dramatic, with the prediction of 60% of over-65-year-olds living in an Asian country by 2030. East Asia leads the demographic transition in Asia, and by 2050, almost all SEA Region Member States will have 20% to 30% of their total populations aged 60 years or over [3].

The fact that people are living longer with chronic diseases represents a positive development overall, but it brings numerous challenges to public health professionals and policymakers. Living longer does not necessarily mean living healthier lives, and some diseases that were deadly in the past, reducing life expectancy, may now be medically managed, despite their impact on individuals’ quality of life [4,5].

In terms of oral health, an increase in tooth retention has increased the prevalence of oral diseases, and as a consequence, the treatment of older patients has changed dramatically over the past years. Advances in dental materials and techniques, the widespread use of fluorides, and better widespread knowledge on prevention are the main reasons for this increase in tooth retention, but caries remain a major concern in older age groups [6]. Teeth are being restored and maintained into old age as opposed to being extracted, leaving them exposed to a number of risk factors. These include xerostomia, caused by several medications usually taken by older people, and exposed root surfaces as a result of gingival recession, poor diet, and partial denture wear [7,8]. Other oral health problems commonly seen among older adults are periodontal disease and oral cancers [9].

In general, the prevalence of oral diseases remains high worldwide, and despite differences among countries, untreated caries in permanent teeth was the most prevalent condition in all of GBD 2015. The global burden of disease study showed that although peaking at the 15- to 19-year-old age group, caries on permanent teeth affected approximately 35% of 65-year-olds. The prevalence of total tooth loss peaked at 75 to 79 years, while that of severe periodontal disease peaked nearly two decades earlier [10].

Therefore, whereas in the past, treating older patients was restricted to denture provision, now, more and more patients are presenting to their dentists with a high demand for treatment [11].

As older individuals carry a lifelong history of dental disease and dental treatment, managing a failing dentition in older individuals may be a complex undertaking for dentists. Treatment decisions may become more complicated when dealing with frail and semi-dependent and dependent patients. Whereas general dentists may be comfortable treating healthy older adults, some studies have suggested that they do not feel prepared to deal with patients that present with a number of comorbidities and polypharmacy and are in need of more tailored (special) oral healthcare [12,13].

Geriatric dentistry or gerodontology refers to the oral healthcare provision to older adults with one or more chronic debilitating illnesses [14].

The European College of Gerodontology (ECG) has published guidelines that highlight the need to modify undergraduate and postgraduate curricula in order to appropriately prepare dentists to deal with the increasing number of older patients [15]. As geriatric dentistry encompasses a much broader concept than mere tooth-loss management and dental treatment for older adults, a new graduate is now required to understand the impact of health status on clinical decision-making. They should also understand that this has a substantial impact on the type of care provided for the elderly. A new graduate also needs to understand that many elderly patients cannot cope with extensive dental procedures, nor do they have the ability to maintain complex restorative treatment. Furthermore, there are a number of communication challenges with frail elderly patients, and securing consent for care often involves the participation of the patient’s family and carers. Accordingly, the new graduate will have to develop insights into these issues and how to manage them appropriately.

Currently, the undergraduate training in gerodontology offered by dental schools seems to be limited, with great variation among dental schools worldwide [16,17].

While the didactic teaching of gerodontology appears to be addressed in undergraduate curricula, a number of experts have recommended greater exposure to clinical management at the undergraduate level. In particular, they highlight the need to train undergraduates to manage frail older adults and provide greater opportunities for inter-disciplinary care [16,17,18].

Therefore, this article aims to explore the challenges in treating older adults and the gaps that currently exist in dental training to meet the demand of an increasingly dentate older population. It also recommends a training pathway to prepare the dental workforce for dealing with the diversity of older individuals’ care needs.

### Heterogeneity of Older Groups

Undergraduate programmes, by and large, train dentists to provide basic primary care dentistry, and this is reflected in the ADEE competency framework [19]. However, the group of older patients is quite diverse, and different groups of elderly may present with care needs that go beyond primary dental care.

The definition of old is usually based on retirement age, which is 65 in many countries. Younger old, or those aged 65–74, years tend to be relatively healthy and active; people aged 75–84 years are the old or mid-old, and their health status can vary from being healthy to those carrying a number of chronic diseases. The oldest old are those 85 years+ and tend to be physically frailer [20]. However, general health status may vary substantially among people over 65, and using chronological age to identify older individuals may not be appropriate. Older adults may be robust or frail, independent or dependent, community-living or institutionalised, and this will have major implications on their treatment needs and pathways [21]. Regarding oral health, such variation can also be seen among the different age groups, and an increasing population of dentate older individuals means that chronological old age is not necessarily an indicator of treatment need or treatment type. Therefore, age in itself should not determine care pathways, and patients’ physical, psychological, and social circumstances are important determinants of oral health and outcomes of treatment [21,22]. For instance, a patient can be chronologically advanced in years, but living independently and in good health. The type of oral healthcare for this type of patient is going to be quite different when compared to a medically compromised, dependent patient of similar age.

The WHO has defined frailty as “a clinically recognizable state in which the ability of older people to cope with everyday or acute stressors is compromised by an increased vulnerability brought by age-associated declines in physiological reserve and function across multiple organ systems” [23]. Though not a disease in itself, frailty is associated with disability, hospitalisation, institutionalisation, and mortality [24]. Frail older people are therefore at higher risk of oral diseases and might need to be managed differently, as comorbidities and polypharmacy can further complicate dental treatment. In this context, pragmatic care to achieve oral comfort, rather than comprehensive care to restore oral function, may be considered [22,25,26].

The Canadian Study of Health and Aging (CSHA) framework (Table 1) [27] categorises older adults into levels of dependency, and this is useful when planning care, bearing in mind the status of the patient. This model is widely used in studies of frailty, including those on its relationship with oral health [28]. As it considers both medical and social aspects, it may be a useful framework to categorise patients with a view to planning training and service delivery.

## 2. Systemic Diseases and Oral Health 

The prevalence of multimorbidity (the presence of two or more comorbidities) is strongly associated with age and represents one of the greatest challenges in healthcare [29]. The interplay of these diseases and the medications used to treat them may have repercussions in other parts of the body, and this is not different when oral health is concerned. In addition to its impact on quality of life, the interlinks between oral and general health have become more evident in the past decades. Malnutrition, for example, which can result from poor nutrient intake caused by poor chewing capacity or oral discomfort, is one of the most relevant problems affecting general health. Studies have shown an association between functional teeth units and poor nutritional status [30]. Periodontal disease has been linked to a number of diseases such as diabetes, chronic kidney disease, recurrent pneumonia, chronic obstructive pulmonary disease, gastritis, rheumatoid arthritis, cancer, and cognitive impairment [31]. A significant body of evidence suggests that poor oral hygiene/denture hygiene may be a risk factor for aspirational pneumonia among older adults. This is also known as nursing home-acquired pneumonia and is the leading cause of mortality amongst nursing home residents [32]. 

Chronic dental diseases are particularly problematic as older adults move from community living into long-term residential care facilities. These patients are often very frail, and many of them may already have a range of systemic comorbidities including cognitive decline or dementia. The impact of a decline in their oral health can be significant, as they have reduced access to dental care and many might have difficulty maintaining adequate oral hygiene [33]. Furthermore, in older adults with serious illness and functional dependency, these oral health problems are often underdiagnosed and untreated [34]. Polypharmacy can also impact oral health and dental treatment, making the management of oral disease in these patients very challenging for dental practitioners. Therefore, when providing dental treatment to older patients, their general health status and drug regime are of major concern [35]. The presence of multimorbidity and frailty may create a demand for more specialised oral healthcare, as dentists treating these individuals must understand the special needs of older people and their ability to undergo and respond to care. Dentists treating this group of older patients should also be prepared to work closely with the multidisciplinary team involved in the patient’s care.

## 3. Barriers to Oral Healthcare

Older individuals face a number of barriers to accessing dental care, which could be classified as patient-related, dentist-related, or general. Patient-related barriers include financial limitations, communication issues, difficulty getting to the dentist, low levels of oral health literacy [36], and non-seeking dental care behaviour. The latter could be due to, among other things, not having oral health as a priority and acceptance of tooth loss as a natural life process [37,38].

Frail individuals may face further barriers as they become dependent on other people to organise oral care for them. They may be hospitalised or living in long-term care facilities (LTCFs), where pathways of oral care may be non-existent. Some of these patients may already have a range of systemic comorbidities such as dementia, which can also impact their oral health, as keeping oral hygiene may become challenging. The result is high levels of dental disease among these patients [39] and a lack of proper oral healthcare pathways in most countries [40]. As a consequence of this erratic pattern of dental treatment, dental extractions may become the only feasible treatment option. 

In Ireland, for example, anecdotal evidence suggests that presently, dental services for LTCF residents are almost exclusively limited to emergency care and that preventative interventions within these facilities are severely lacking. Some practitioners may agree to deliver dental treatment in LTCFs out of their own goodwill, and the number of dentists to undertake such a task is limited [41].

Some dentist-related barriers may deter the organisation of dental treatment to frail and semi-dependent or dependent older patients, and these include a lack of adequate equipment to see patients outside practice, a lack of adequate reimbursement, and a lack of training and experience [12,35]. These have been mainly associated with the treatment of older individuals in LTCFs but also in private practice. In a study conducted in the Netherlands, dentists have agreed that treating frail older patients in practice is very challenging and that even if they decide not to treat these patients, it is difficult to find an experienced professional to refer them to. Inadequate reimbursement was also mentioned by the majority of dentists in this study as a deterrent to treating frail elderly [42]. 

The problem can be further complicated by the complex nature of the dental procedures usually required by this patient cohort. Increased levels of tooth retention have resulted in a greater variety of restorative work being delivered to adults, such as crowns, bridges, and implants. Like any other dental work/material, these have a limited lifespan and will need maintenance and eventual repair or replacement. The problem is that these failures might occur later in life when individuals present with systemic disease and polypharmacy and might be frail and incapable of maintaining oral hygiene. The result is that clinicians treating older individuals may be constantly facing difficult treatment decisions that require not only knowledge of medical conditions affecting older patients and their oral health and treatment management strategies but also some clinical experience [43].

## 4. Gerodontology Training

The need to develop training in gerodontology has been recognised internationally, and the European College of Gerodontology (ECG) has developed undergraduate curriculum guidelines in order to respond to the treatment needs of a growing elderly population [15]. At the postgraduate level, training in gerodontology is offered by a number of schools in Europe, but this is usually embedded in other speciality programs, such as prosthodontics. Postgraduate courses exclusively dedicated to gerodontology seem scarce [44].

In Europe, most schools offer teaching in gerodontology at the undergraduate level, but this is mostly didactic. Clinical training is usually embedded in other disciplines’ clinics [45]. This means that students may acquire a basic understanding of the complexities of treating the different older age cohorts, but have no direct clinical experience to carry out procedures independently. The ECG suggests that in order to achieve adequate training at an undergraduate level, clinical training, particularly for the frail and dependent elderly, could be placed in the final year of training, after students have been exposed to the main areas of dentistry and general medicine. However, there are several barriers to implementing a gerodontology curriculum at the undergraduate level, as recommended by the ECG guidelines. These include a busy curriculum and a lack of trained staff and resources in most dental schools [45]. The nature of care for elderly patients varies according to health status and social and economic circumstances. For many elderly, it is not straightforward to come to dental hospital clinics for care. Additionally, students do not gain “real world” experience when all of their training is undertaken in a hospital-based setting. For these reasons, it is highly desirable to provide undergraduate training under supervision in a community-based setting as well as in dental hospital clinics. 

Timely access to oral healthcare for those with multi-morbid chronic conditions is problematic for many older adults [44]. This is likely to be influenced by primary dental care practitioners’ willingness to treat such patients if they do not feel adequately trained in managing older adults with special needs. If the training of undergraduates and new graduates included comprehensive treatment planning and provision for medically compromised older adults and those in residential care settings, then it is likely that access to appropriate care for them would be enhanced. 

Given the multiple challenges in providing care for medically compromised older adults with special needs, it is highly unlikely that all new graduates from dental schools can be deemed competent in gerodontology. Perhaps a more realistic view of this is for undergraduate education to provide a suitable foundation to be developed further in postgraduate education. Figure 1 shows a model of care for older adults according to the level of dependency on the CHSA scale [13]. In this model, it is proposed that older adults classed as levels 1–3 (no dependency–pre-dependency) have care needs that can be managed by general dental practitioners in primary dental care settings. If they are classed as dependency levels 4 or 5 (i.e., low or medium dependency), then it is likely that they will have care needs that are beyond the scope of the practice of general dental practitioners who have not had additional training in special care dentistry for older adults. Furthermore, the care setting may need to be either domiciliary (i.e., in the patient’s home) or a suitably equipped community care facility.

To upskill general dental practitioners to care for these patients, postgraduate certification programmes could be structured to provide such training. This could be along the lines of micro-credentialled courses with didactic and clinical components or even advanced diploma-level qualifications. The clinical component should ideally include training in domiciliary and/or community-based settings. With this training, the generalist will have advanced-level clinical skills. Finally, if the patient is classed as level 6 or 7 (high dependency), then it is likely that the management of such patients will be complex. This will include patients with multiple co-morbidities and moderate-to-advanced cognitive impairment. The care of these patients will almost certainly require multi-disciplinary input and in many cases will require management in hospitals or special care facilities. The proficient management of these patients will require a dentist to have significant additional training and experience and professional working relationships with relevant medical and allied health professional teams (e.g., geriatric psychiatry, neurology, and rehabilitation specialist teams).

Figure 2 is a proposed framework to illustrate the training/education continuum from undergraduate education to expert status. This framework aligns the level of training and competence to the care needs of older patients and the care setting.

For some older adults, particularly as they become or are “functionally dependent” or have moderate-to-advanced cognitive decline, oral care should ideally be integrated into a holistic care plan. This involves multi-disciplinary care input, and a “Team” approach would be ideal. Figure 3 indicates what this might look like in a patient-centric care model. Basic care for functionally independent older adults can be provided in a primary care setting, but as the patient becomes frail and dependent, the need for team-based care increases. The dental expertise required is more specialised, and oral care needs to be coordinated with input from medical care and allied health professionals. Ideally, care protocols would be locally developed and would clearly identify referral pathways for medically compromised patients to access appropriate care as quickly as possible. To date, there are no internationally accepted dental care guidelines for medically compromised older adults, and this should be addressed.

## 5. Training of Non-Dental Healthcare Professionals

The development of geriatric oral health educational programmes for non-dental healthcare workers has been recommended by several experts in the field of gerodontology [15]. This recognises the need for team-based care for older adults with special care needs. They have suggested training professionals at all levels of education (undergraduate, postgraduate, and speciality training), including physicians, nurses, nursing assistants, physiotherapists, occupational therapists, medical assistants, pharmacists, dieticians, and others.

One of the reasons for this is that despite the high prevalence of oral diseases among older individuals, dental attendance among these individuals is low, and whereas they may not see a dentist routinely, other health professionals are involved in their treatment and care, sometimes on a daily basis.

Therefore, these non-dental health workers could potentially provide some form of oral health counselling and oral assessment, perform basic preventive measures, and refer the patient to a dentist or hygienist when appropriate. The ECG has developed a list of learning objectives for training non-dental healthcare professionals in the oral health assessment and promotion of older adults, and this includes performing an initial assessment of oral health status, demonstrating oral hygiene measures to older adults and their caregivers, and assisting with or providing daily oral hygiene when necessary [46].

Training non-dental healthcare professionals in gerodontology becomes particularly important when we consider residents of LTCFs, who historically present with poor oral health and oral hygiene [47,48]. These elderly usually depend on caregivers to perform some daily activities, such as toothbrushing and cleaning their dentures, and despite the good intentions of nurses and caregivers to do so, the lack of knowledge and skills and a negative attitude towards oral healthcare seem to hinder the staff’s ability to perform these routine activities [49].

Several studies have shown that education programmes aimed at nurses can improve their knowledge of oral health [50]. However, these changes seem to be short-term and not followed by a change in attitude. Additionally, although oral healthcare protocols may be in place, the oral healthcare provided in LTCFs does not seem to comply with the available guidelines and protocols [51]. This points to the need to also establish clear evidence-based implementation strategies to improve oral healthcare in these institutions.

The introduction of oral care aides in LTCFs has also been shown to positively impact the residents’ oral health and the staff’s experience of oral healthcare. Training aides to conduct oral health assessments, communicate with dental professionals, inform staff of oral healthcare issues, and document care plans could improve residents’ oral health [52,53].

## 6. Conclusions

Older individuals are a diverse group with different oral care needs. Providing oral healthcare to the different elderly groups calls for dental staff with training levels and competence aligned with the care needs of these different groups and care settings. Whereas the principle of having greater integration of gerodontology in training programmes appears to be accepted, further research may be required to test new models of training and how these could impact patient care for an ageing population. Furthermore, there is a need for structured engagement both locally and globally between all the stakeholders (i.e., dental school representatives, patient representatives, and policymakers) to map out options for service delivery to this growing population with diverse needs.

## Figures and Tables

**Figure 1 ijerph-20-06234-f001:**
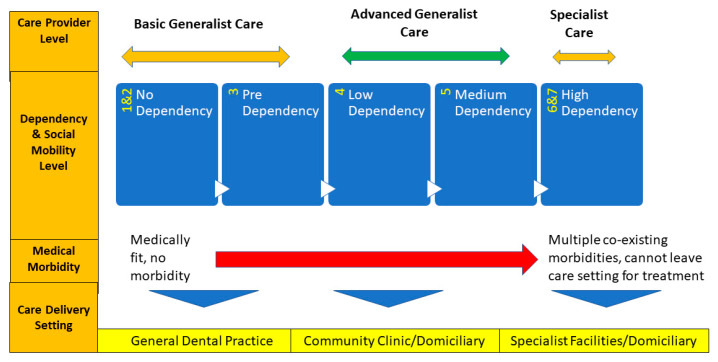
Care delivery model for older adults according to medical status, level of dependency, and social mobility level. The model is underpinned by the right siting of care principle.

**Figure 2 ijerph-20-06234-f002:**
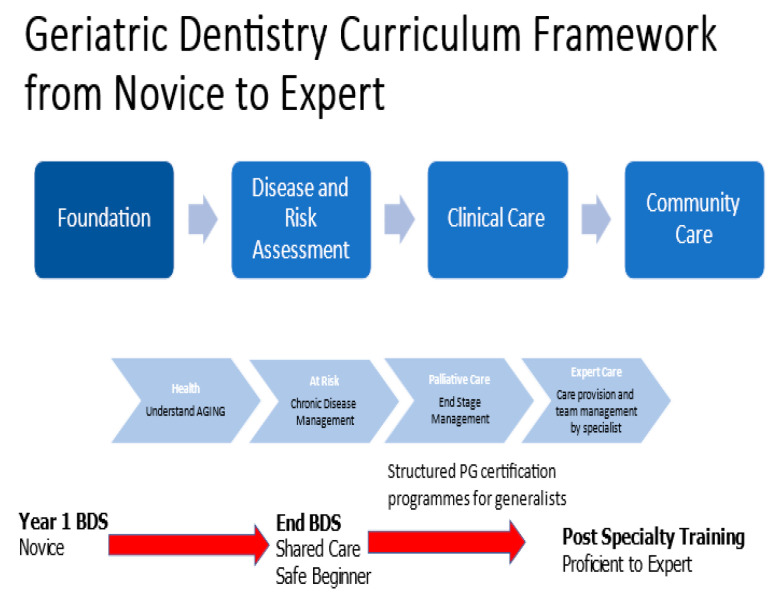
Proposed education/training continuum for geriatric dentistry.

**Figure 3 ijerph-20-06234-f003:**
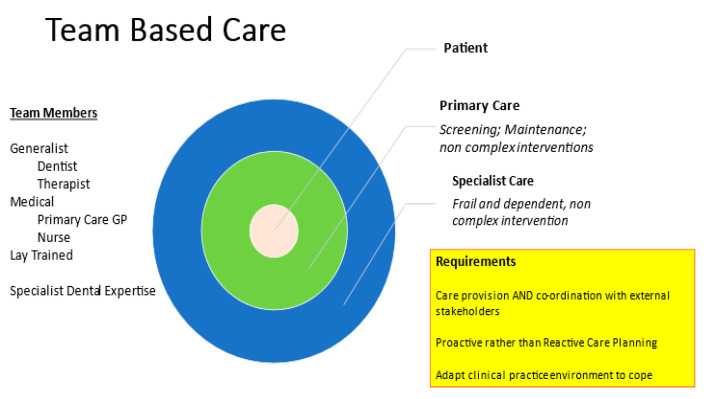
Team-based, patient-centric care concept for robust and medically compromised older adults.

**Table 1 ijerph-20-06234-t001:** The CSHA dependency framework (adapted from “Oral health: caring for older adults”. FDI, 2019).

Level of Dependency	Definition
No dependency, CSHA levels 1 and 2	Robust people who exercise regularly and are the fittest group for their age.
Pre-dependency, CSHA level 3	People with chronic systemic conditions that could impact oral health but, at the point of presentation, are not currently impacting oral health. A comorbidity whose symptoms are well controlled.
Low dependency, CSHA level 4	People with identified chronic conditions that are affecting oral health but who currently receive or do not require help to access dental services or maintain oral health. These patients are not entirely dependent, but their disease symptoms are affecting them.
Medium dependency, CSHA level 5	People with an identified chronic systemic condition that currently impacts their oral health and who receive or do not require help to access dental services or maintain oral health. This category includes patients who demand to be seen at home or who do not have transport to a dental clinic.
High dependency, CSHA levels 6 and 7	People who have complex medical problems preventing them from moving to receive dental care at a dental clinic. They differ from patients categorised as medium dependency because they cannot be moved and must be seen at home

## Data Availability

Not applicable.

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
