# Peer review of "Providing Oral Healthcare to Older Patients—Do We Have What It Takes?"

_ijerph, 2023, doi:10.3390/ijerph20136234_

Round 1
Reviewer 1 Report
I would like to appreciate the authors for their efforts and contribution to this review. I hereby give a few suggestions for a better and broader reach to the readers.
Introduction:
i.The authors can briefly explain the most common oral conditions that are seen among the older population.
ii. I strongly suggest the authors provide a table that illustrates the prevalence of oral conditions among older populations around the globe.
iii. Line 51-52, I suggest the authors justify the statement by providing additional information regarding the limited training in gerodontology offered by dental school.
iv. Line 66-68, the justification of this review needs to be stronger so that it serves the purpose.
Methodology: (Please include a sub-section)
i. I would like to suggest the authors provide the flow of extracted information while writing this review.
For eg: From which source the information has been extracted?
* If possible through a flowchart.
Discussion: (Please include a sub-section)
i. Please elaborate on the advantages, benefits, and barriers of having gerodontology training.
Conclusion:
i. Conclusions of the review need to be elaborated as it does not provide enough justification for this review.
Moderate English edit and spell checks needed.
Author Response
The authors would like to thank the reviewers’ for their suggestions to improve the manuscript. Please find below a response to each of the reviewer’s comments.
i.The authors can briefly explain the most common oral conditions that are seen among the older population.
Response: Information has been added to the Introduction, lines 53- 69.
- I strongly suggest the authors provide a table that illustrates the prevalence of oral conditions among older populations around the globe.
Response: Thank you for your suggestion. Because data on prevalence of oral conditions vary worldwide, and is usually collectively reported, the authors believe that using data from the Global Burden of Diseases study may be the best approach here. Text was added to Page 2, lines 64 to 69, as there was not sufficient information for a table.
iii. Line 51-52, I suggest the authors justify the statement by providing additional information regarding the limited training in gerodontology offered by dental school.
Response:Thanks for this. Text has been added on Page 3, lines 99 -103.
The limited training in Geroontology is further discussed on Page 6, lines 226-241.
- Line 66-68, the justification of this review needs to be stronger so that it serves the purpose.
Response: The justification for the review is further enhanced on Page 3, lines 97-107.
Methodology: (Please include a sub-section)
- I would like to suggest the authors provide the flow of extracted information while writing this review.
For eg: From which source the information has been extracted?
* If possible through a flowchart.
Response: Thanks for this suggestion. However, the authors would like to clarify that this is not meant to be a systematic review, but a commentary paper. For this reason, a narrative of available data on the various aspects of dental care provision to older patients was undertaken, and for this reason, the usual sections Methods and Results are absent.
Discussion: (Please include a sub-section)
- Please elaborate on the advantages, benefits, and barriers of having gerodontology training.
Response: Barriers are covered on Page 6, lines 234-239., but further text has been added to clarify this Page 6 lines 243-249.
Conclusion:
- Conclusions of the review need to be elaborated as it does not provide enough justification for this review.
Response: Thanks for this comment. Text has been added to the conclusions.
Reviewer 2 Report
Peer Review Report
This is an interesting review that addressed oral health care for older patients. While the manuscript is well writing and adds to the literature. There are some points that should be addressed to enhance the overall quality of the manuscript.
Abstract:
The abstract needs improvement as it currently lacks specific findings of the review. It reads more like a rationale rather than a summary of the review's outcomes. Please provide a more detailed overview of the findings and their implications.
Introduction:
P1 L29-31: "In Asia, the statistics are even more dramatic, with the prediction of 60% of over 65-year-olds living in an Asian country by 2030 [3]." This sentence would benefit from being more specific. Please provide additional details or specify the Asian countries involved.
P1 L32-36: Please include a reference to support the information presented in this sentence.
P1 L37-41: It would be useful to briefly introduce the reasons behind the changes in oral health during older age and why people are now maintaining dentition into old age.
P2 L71-88: To support the discussion on frailty, robustness, and dependency related to older populations, please provide more references. These aspects are crucial in understanding the challenges faced by older individuals.
P3 L92: It would be helpful to introduce the term "frailty" in this context, as it is one of the most profound implications of aging.
L99: Since there are various models and tools for measuring dependency, such as ADL and IADL, please clarify why this particular model was chosen for the study. Address this point before presenting the table. Additionally, has this model been tested in dental settings before? Please provide relevant information.
Conclusion: Please discuss the public health and practice implications of the study in this section. Highlight how the findings can be applied to real-world situations and provide recommendations for oral healthcare providers.
Overall, the manuscript requires revisions to address the specific points mentioned above.
Author Response
The authors would like to thank the reviewers’ for their suggestions to improve the manuscript. Please find below a response to each of the reviewer’s comments.
Abstract:
The abstract needs improvement as it currently lacks specific findings of the review. It reads more like a rationale rather than a summary of the review's outcomes. Please provide a more detailed overview of the findings and their implications.
Response: Thanks for this suggestion. The abstract has been amended accordingly.
Introduction:
P1 L29-31: "In Asia, the statistics are even more dramatic, with the prediction of 60% of over 65-year-olds living in an Asian country by 2030 [3]." This sentence would benefit from being more specific. Please provide additional details or specify the Asian countries involved.
Response: Information has been added: Lines 45-47.
P1 L32-36: Please include a reference to support the information presented in this sentence.
Response: References 4 and 5 have been added.
P1 L37-41: It would be useful to briefly introduce the reasons behind the changes in oral health during older age and why people are now maintaining dentition into old age.
Response: Information has been added on Page 2 lines 55-60.
P2 L71-88: To support the discussion on frailty, robustness, and dependency related to older populations, please provide more references. These aspects are crucial in understanding the challenges faced by older individuals.
Response: References have been added on Heterogeneity of older groups, Pages 3 and 4,
P3 L92: It would be helpful to introduce the term "frailty" in this context, as it is one of the most profound implications of aging.
Response: Information has been added on page 3 and 4, lines 131-135.
L99: Since there are various models and tools for measuring dependency, such as ADL and IADL, please clarify why this particular model was chosen for the study. Address this point before presenting the table. Additionally, has this model been tested in dental settings before? Please provide relevant information.
Response: Information has been added on page 4, lines 142-145.
Conclusion: Please discuss the public health and practice implications of the study in this section. Highlight how the findings can be applied to real-world situations and provide recommendations for oral healthcare providers.
Response: Information has been added to the Conclusions, lines 342-348